# Protease–Antiprotease Imbalance in Bronchiectasis

**DOI:** 10.3390/ijms22115996

**Published:** 2021-06-01

**Authors:** Martina Oriano, Francesco Amati, Andrea Gramegna, Anthony De Soyza, Marco Mantero, Oriol Sibila, Sanjay H. Chotirmall, Antonio Voza, Paola Marchisio, Francesco Blasi, Stefano Aliberti

**Affiliations:** 1Respiratory Unit and Cystic Fibrosis Adult Center, Fondazione IRCCS Ca’ Granda Ospedale Maggiore Policlinico, 20122 Milan, Italy; martina.oriano@unimi.it (M.O.); francesco.aamati@gmail.com (F.A.); gramegna.med@gmail.com (A.G.); marco.mantero@unimi.it (M.M.); paola.marchisio@unimi.it (P.M.); francesco.blasi@unimi.it (F.B.); 2Department of Pathophysiology and Transplantation, Università degli Studi di Milano, 20122 Milan, Italy; 3Population and Health Science Institute, NIHR Biomedical Research Centre for Ageing & Freeman Hospital, Newcastle University, Newcastle NE2 4HH, UK; anthony.de-soyza@newcastle.ac.uk; 4Respiratory Department, Hospital Clinic, IDIBAPS, CIBERES, 08036 Barcelona, Spain; osibila@clinic.cat; 5Lee Kong Chian School of Medicine, Nanyang Technological University, Singapore 639798, Singapore; schotirmall@ntu.edu.sg; 6Emergency Department, IRCCS Humanitas Research Teaching Hospital, 20122 Milan, Italy; antonio.voza@humanitas.it; 7Paediatric Highly Intensive Care Unit, Fondazione IRCCS Ca’ Granda Ospedale Maggiore Policlinico, 20122 Milan, Italy

**Keywords:** bronchiectasis, proteases, neutrophilic inflammation

## Abstract

Airway inflammation plays a central role in bronchiectasis. Protease–antiprotease balance is crucial in bronchiectasis pathophysiology and increased presence of unopposed proteases activity may contribute to bronchiectasis onset and progression. Proteases’ over-reactivity and antiprotease deficiency may have a role in increasing inflammation in bronchiectasis airways and may lead to extracellular matrix degradation and tissue damage. Imbalances in serine proteases and matrix-metallo proteinases (MMPs) have been associated to bronchiectasis. Active neutrophil elastase has been associated with disease severity and poor long-term outcomes in this disease. Moreover, high levels of MMPs have been associated with radiological and disease severity. Finally, severe deficiency of α1-antitrypsin (AAT), as PiSZ and PiZZ (proteinase inhibitor SZ and ZZ) phenotype, have been associated with bronchiectasis development. Several treatments are under study to reduce protease activity in lungs. Molecules to inhibit neutrophil elastase activity have been developed in both oral or inhaled form, along with compounds inhibiting dipeptydil-peptidase 1, enzyme responsible for the activation of serine proteases. Finally, supplementation with AAT is in use for patients with severe deficiency. The identification of different targets of therapy within the protease–antiprotease balance contributes to a precision medicine approach in bronchiectasis and eventually interrupts and disrupts the vicious vortex which characterizes the disease.

## 1. Introduction

Bronchiectasis is a chronic respiratory disease characterized by an irreversible pathological dilation of the bronchi associated with a chronic syndrome of cough, sputum production and recurrent respiratory infections [1]. Bronchiectasis prevalence and incidence are increasing worldwide [2]. The highest prevalence and incidence of this disease have been reported in the UK with a prevalence in 2013 of 566.1 per 100,000 in women and 485.5 per 100,000 in men and an incidence of 35.2 per 100,000 person-years in women and 26.9 per 100,000 person-years in men [3]. Other data report an incidence of 362 patients per 100,000 population and a prevalence of 48.1 person-years per 100,000 in 2012 in Catalonia and 163 per 100,000 population, with an incidence of 16.3 person-years in 2020 in Italy [2,4]. Bronchiectasis can be caused by several different genetic and acquired conditions [5]. In terms of pathophysiology, one of the current paradigms for bronchiectasis onset and progression is represented by a vicious vortex in which bacterial infection, airway inflammation, lung tissue disruption, and impaired mucous clearance are regarded as the major components [6,7]. Each of those may be the entry point of the vortex, which eventually leads to bronchiectasis progression [8]. For instance, infection with non-tuberculous mycobacteria (NTM) induces a local inflammatory status which determines parenchymal structural damage [9,10]. Ciliary dysfunction and failure of the muco-ciliary clearance, as demonstrated in patients with primary ciliary dyskinesia (PCD) and cystic fibrosis (CF), increases the risk of pulmonary infections and airway inflammation leading to the chronicity of the vortex [11,12,13]. Systemic immune diseases, such as immunodeficiency or extra-pulmonary autoimmune diseases, may cause a delayed resolution of respiratory infections followed by inflammation and structural lung damage, determining the entry in the pathophysiological vicious cycle [14,15].

Airway inflammation plays a central role in chronic respiratory diseases and, especially, in bronchiectasis [16,17,18]. The vicious vortex has been identified as the target of treatments in bronchiectasis and several approaches were adopted or are in study to target inflammation in this disease. Inflammatory biomarkers were associated with disease severity, clinical outcomes and are nowadays targets of treatment [16,17,18]. Proteases are molecules with several mechanisms of action in bronchiectasis, ranging from fighting infections, regulating inflammation and being involved in tissue remodeling [19,20]. Alterations in protease–antiprotease balance have been associated with bronchiectasis and are nowadays considered as targets of treatment [17,21,22]. Although also bacteria may produce proteases that may be involved in the protease–antiproteases balance in bronchiectasis, we decided to focus only on proteases produced by the host [23,24].

## 2. Search Strategy and Selection Criteria

We searched PubMed for articles published in or translated into the English language between 1 January 2010 and 1 April 2021, using combinations of the following terms: “proteases”, “bronchiectasis”, “treatable traits”, “personalised medicine”, “proteases treatment”, “antiproteases”, “lungs”. We determined relevance based on content. We also found articles through authors’ personal files and from references cited in retrieved articles. The final reference list was generated based on relevance to this personal view.

## 3. Airway Inflammation in Bronchiectasis

Neutrophilic inflammation represents the major response to infectious triggers in bronchiectasis airways [25]. Neutrophils are abundant in bronchiectasis airways, and neutrophilic inflammatory effectors are frequently secreted by these cells, thus contributing to the inflammatory status and suppressing external threats [26]. Several enzymes are associated with neutrophilic inflammation in bronchiectasis, and a high number of proteins coordinate neutrophil recruitment into the lung [27]. Once in the lungs, neutrophils hinder microbial infection through the secretion of several effectors, including cathelicidins, myeloperoxidases (MPO), serin proteases, lactotransferrin and cytokines, interleukin (IL)-1α, IL-1β, tumour necrosis factor (TNF)-α, IL-8, IL-12β and others [26,28,29]. Imbalances in cytokines interplay in bronchiectasis airways are associated with both disease severity, as in the case of IL-1β, and worse radiological involvement, as in the case of IL-8 [30,31]. Furthermore, neutrophils form extracellular traps (NETs) through the extrusion of chromatin DNA, histones and bactericidal proteins in order to actively kill bacteria. A recent study has confirmed through proteomics analysis the high presence of NETs proteins in sputum and their strong association with disease severity [32]. Another recent study has also demonstrated the presence of bactericidal proteins such as LL-37, a cathelicidin-derived peptide with a broad spectrum of antimicrobial activity, in sputum samples of bronchiectasis patients [33]. The balance between inflammatory and anti-inflammatory cytokines is one of the determinants of disease status in bronchiectasis, and a dysregulated signaling may perpetuate the inflammatory status [34,35].

However, up to one third of bronchiectasis patients might have an eosinophilic immune response [36,37,38,39]. Recent experiences focused on the importance of the identification of eosinophilic inflammation as a treatable trait in bronchiectasis, underlying the role of inflammatory response in bronchiectasis [39,40]. Proteases may also have a role in the eosinophilic response, and further studies will be needed to unravel this aspect of bronchiectasis pathophysiology.

## 4. An Overview on Proteases and Antiproteases in the Airways

The balance between different inflammatory effectors in the airways is fundamental in bronchiectasis, and its disruption may be associated with disease severity and progression [34,35]. Among inflammatory effectors contributing to infection resolution in bronchiectasis, proteases play the central role to directly fight micro-organisms invasion and regulate other inflammatory effectors. Both proteases and antiproteases regulate physiological processes, including regeneration, repair and fighting local infections [41]. High levels of proteases, as well as decreased production of antiproteases, are involved in the pathophysiology of bronchiectasis and contribute to the onset and the sustaining of the vicious vortex. Several experiences reported an imbalance in serine proteases, matrix metalloproteinases and cysteine proteases in bronchiectasis with an associated pulmonary dysfunction [17,18,20,42].

### 4.1. Serine Proteases

Neutrophil serine proteases, which belong to the chymotrypsin family, are stocked in an activated form in the azurophil granules of neutrophils and secreted upon noxious stimuli in the lungs. Neutrophil elastase (NE), cathepsin G (Cat-G) and proteinase 3 (PR3) are the main serine proteases. They are produced as zymogens during neutrophilic differentiation and activated before or during transports to granules cathepsin C, also known as dipeptidyl peptidase 1 (DPP1). These proteases are involved in the non-oxidative pathway of intracellular and extracellular pathogen destruction both in free form attached to the cellular membrane and in NETs. Intracellularly serine proteases support the digestion of phagocyted microorganisms within phagolysosomes [41].

In the extracellular environment serine proteases degrade bacterial virulence factors [41]. They are released extracellularly in active form, and are able to bind and trim bacterial flagellin, depolarize bacterial membranes, inhibit protein synthesis, activate growth factors through proteolysis, cleave adhesion molecules and contribute to lymphocyte activation [43].

These enzymes also have a role in the activation/inactivation of cytokines and chemokines. They take part in the regulation of both production and activation of IL-8 along with TNFα, and IL-1β, directly or through the interaction with specific receptors (TLR or PAR) which are able to initiate transcription cascade. NE is also responsible for the activation of other inflammation effectors (Figure 1) [41].

Cat-G clears pathogens, modifies chemokines and cytokines and, thus, regulates inflammation [44]. Interestingly, Cat-G have been associated to poor *P. aeruginosa* clearance in CF mouse airways [45].

PR3 is an enzyme able to cleave structural proteins for tissue remodelling, to regulate immune response to bacterial triggers through cleavage of antibacterial peptides, activation of pro-inflammatory cytokines and regulation of cellular processes. PR3 is also active in cleaving C1 inhibitor, IL-8, TNF, IL-1, TGF [46].

#### Neutrophil Elastase

NE is a 218-amino-acids long protein coded in ELANE (Elastase, Neutrophil Expressed) gene and is the most abundant and studied serine protease [44]. NE may have both an intracellular and an extracellular mechanism of action, and the extracellular NE may exert its function as membrane bound or soluble protein [44]. Soluble NE action is further regulated by the presence of different inhibitors including α1 antitrypsin (AAT).

Serine proteases and specifically NE secretion have been associated with chronic infection with Gram-negative bacteria, including *Pseudomonas aeruginosa* [44]. Experiments conducted on mice suffering from *P. aeruginosa*-induced pneumonia demonstrated that the absence of NE was associated with decreased levels of pro-inflammatory cytokines, including TNF-α, macrophage inflammatory protein-2 (MIP-2), and IL-6 in the lungs. NE together with the modulation of cytokine expression contributes to the host protection against *P. aeruginosa* [47]. Although NE and proteases in general have a physiological and beneficial role in lungs, high levels of this protease may cause tissue damage, increased mucus production, decreased ciliary beating rate, and enhanced lung epithelium damage [48]. NE-dependent structural damage may lead to irreversible airway dilation and, thus, the development of bronchiectasis [49,50]. Antiproteases also have a major role in regulating proteases and hence their deficiency may contribute to the prolonged uncontrolled protease activity leading to subsequent inflammatory responses and tissue damage.

### 4.2. Matrix Metalloproteinases

Matrix metalloproteinases (MMPs) are zinc-dependent proteases which are able to degrade collagen. They are classified based on the specific substrate, and MMP-8 and MMP-9 have been reported as neutrophil MMP [51]. MMPs are synthesized in an inactive form and may be either secreted as pro-proteins or activated intracellularly by convertases. MMPs tissue inhibitors of metalloproteinases (TIMPs) are direct inhibitors of MMPs and, when secreted, are able to bind 1:1 their target and suppress MMPs activity. MMPs are responsible for tissue remodeling and involved in pulmonary immunity [52]. MMPs may act on a large variety of substrates, including inflammatory effectors. For instance, IL-1β, TNFα, IL-8 are cytokines that may be activated or potentiated by MMPs in lungs [49]. MMP-9 is able to cut IL-1β pro-protein and inhibit the activity of IL-1β [50]. IL-8 activity may be enhanced 10 times after MMP-9 activation [53]. MMPs are also involved in the regulation of serine proteases. On one hand, MMP-8 and MMP-9 degrade AAT and restore the activity of inhibited serine proteases, mostly NE [54]. On the other hand, NE inactivates TMP1 and leads to MMP-9 activation [20]. Although MMPs play a crucial role in pulmonary immunity through the activation of defensins and mediation of inter cellular signaling, altered levels of these enzymes may determine a degradation of the extracellular matrix. As mentioned, protease overactivity may affect lung integrity, resulting in bronchiectasis development or contribute to progression.

### 4.3. Cysteine Proteases

Cysteine proteases are part of the papain family and constitutively expressed in many tissues. Cathepsins are part of the cysteine proteases family. Cathepsin C, also known as DPP1, is mostly expressed in myeloid cells and it is responsible for fibronectin and collagens type I, III, and IV cleavage [55,56]. DPP1 has also a role in the activation of serine proteases in neutrophils precursors, and targeting this protease may hinder pathological serine proteases activity [57]. Due to the fundamental role of DPP1 in serine proteases activation, molecules were developed and tested in bronchiectasis in order to treat NE activity in bronchiectasis, with promising results [57]. Other cysteine proteases such as cathepsin S, B or L have been associated with cystic fibrosis (CF) and bronchiectasis severity in CF; however, no data have been published so far in patients with non-CF bronchiectasis [42,56].

### 4.4. Antiproteases

#### 4.4.1. α1 Antitrypsin (AAT)

AAT is an albumin-like antiprotease secreted by hepatocytes, immune cells and bronchial epithelial cells [58]. AAT is able to inactivate through irreversible binding several proteases including serin proteases and it has a high affinity for NE. AAT is an irreversible inhibitor of serine proteases. When AAT docks serine proteases, the reactive central loop is cleaved in a high energy state and it is able to modify its conformation distorting and altering the protease conformation, see Figure 2 [59]. AAT is coded in the SERPINA1 (Serine peptidase inhibitor, clade A member 1) gene and polymorphisms in this gene are associated with functional, dysfunctional, deficient or null variants of the protein and with low levels of AAT in serum. Although more than one hundred polymorphisms in the AAT gene have been reported in literature, the most common variants associated with AAT serum deficiency are Z and S allele. The number of rare variants represents a large variability in AAT production and functionality. These polymorphisms, sometimes with unknown function, may further be associated with AAT deficiency with an increased number of patients who might be affected by this condition [60,61]. Z allele is responsible for a mutated isoform of AAT and has been associated with the most severe deficiency in serum showing around 10% of the serum concentration of the physiological variants. S allele is responsible for a milder deficiency, associated with nearly 60% of physiological AAT concentration in serum [61]. Carriers of pathological variants of AAT proteins with low AAT serum concentration suffer from a rare genetic condition named α1 antitrypsin deficiency (AATD). Patients with Z AAT allele have 5 times less physiological concentration of AAT and increased neutrophilic presence in lungs maybe because of an excess in chemoattractants [62]. A lack of AAT in the lungs may lead to unopposed protease activity and to tissue damage [62].

#### 4.4.2. Antileukoprotease (ALP) Superfamily

The antileukoprotease (ALP) superfamily includes enzymes secreted in the lung that show an inhibitory effect on airway proteases. These proteins are synthesized and secreted locally in the lung in response to primary cytokines such as IL-1 and TNF [63]. This superfamily includes elafin and secretory leukocyte proteinase inhibitor (SLPI).

*SLPI.* SLPI inhibits NE, Cat-G, trypsin, chymotrypsin and chymase, with a high affinity to NE that is its major target. SLPI is believed to have an anti-inflammatory and anti-bacterial role [63].

*Elafin.* Elafin is a protein which inhibits a more limited number of proteases in comparison to SLPI. Its main targets are NE and PR3. In addition to its inhibitory effect, elafin seems to be active against *P. aeruginosa* [63].

## 5. The Role of Proteases and Antiproteases in Bronchiectasis

### 5.1. Neutrophil Elastase in Bronchiectasis

NE represents a key biomarker in bronchiectasis, and levels of active NE (aNE) measured in bronchoalveolar lavage fluid of bronchiectasis patients during stable state are higher compared to healthy controls [64,65]. From a microbiological point of view, NE directly correlates with bacterial load in sputum of patients chronically infected by *P. aeruginosa* [31]. From a clinical point of view, aNE is associated with a higher rate of exacerbations in bronchiectasis. aNE concentration increases during bronchiectasis exacerbations and decreases during and after antibiotic treatment [31,66,67]. Although we are focused on aNE, older methods for NE analysis included non-specific methods, not able to differentiate among serin proteases. In addition, older experiences reported NE concentration instead of activity, along with AAT-NE complex, higher in patients compared to healthy controls [18].

A few years ago, Chalmers and co-workers demonstrated an association between active NE levels in sputum of stable-state bronchiectasis adults and disease severity (evaluated through the Bronchiectasis Severity Index), pulmonary function, dyspnea, radiological severity and poor outcomes during follow-up, including exacerbations and hospitalization [68]. These data have been recently confirmed across two southern European cohorts [18]. Patients were divided into three groups based on concentrations of aNE in sputum (low aNE as aNE < 7 µg/mL, medium aNE between 7 and 20 µg/mL and high above 20 µg/mL). aNE concentration directly correlated with disease severity, and it is inversely associated with quality of life [17,18,68]. The rate of patients with poor lung function was higher in the medium and in the high aNE groups compared to the low aNE ones. aNE levels were also higher in patients with chronic infection especially caused by *P. aeruginosa* [17]. A point of care assay has been recently developed to evaluate aNE in sputum and tested in bronchiectasis patients [16]. This test showed a good performance in identifying in few minutes patients at higher risk of airway infection and exacerbations. Finally, NE has been progressively identified as a relevant target for future treatments in bronchiectasis with pharmaceutical companies working on molecules able to inhibit this protease [57].

NE action in degrading elastin produces peptides including desmosine and isodesmosine, which are molecules frequently measured in serum and urine to evaluate NE activity. Circulating desmosine directly correlated with dyspnea, quality of life, radiological involvement and disease severity in bronchiectasis, and inversely with FEV_1_ (Forced Expiratory Volume in the 1st second) [21]. Moreover, serum desmosine concentration also correlated with sputum concentration of aNE [21]. Finally, high concentration of serum desmosine was also associated with all-cause mortality, respiratory death in the first 3 years, and cardiovascular death after 3 years of follow-up in bronchiectasis [69].

### 5.2. Matrix Metalloproteinases in Bronchiectasis

Imbalances in MMPs levels as well as MMPs-TMPs ratios have been identified in bronchiectasis [70]. Several MMPs are associated with bronchiectasis. Taylor and colleagues reported increased values of MMP-1, MMP-3, MMP-7, MMP-8, and MMP-9 and TIMP-2 and -4 levels, as well as MMP-8/TIMP-1 and MMP-9/TIMP-1 ratios in patients vs. healthy controls: Moreover, MMP-2 and MMP-8 are increased in patients with *H. influenzae*-dominated microbiota compared to those dominated by *P. aeruginosa* [70]. Guan and colleagues reported increased levels of MMP-8, MMP-9 and MMP-9/TIMP-1 ratio in patients with bronchiectasis and found an association between those levels and both disease and radiological severity. Notably, they also found MMP-9 levels increasing during exacerbations [70]. Similar data were found in patients with bronchiectasis in general and in those with primary ciliary dyskinesia with researchers showing an association between high levels of MMPs and poor lung function [71].

### 5.3. Airway Proteases and Microbial Community in Bronchiectasis

The evaluation of airway inflammation alone in bronchiectasis limits our understanding of disease pathophysiology. A recent study evaluated the interaction between microbial community and inflammatory biomarkers [15]. Among 185 adult patients with stable-state bronchiectasis enrolled in a cross-sectional study, those with aNE≥20 µg/mL had higher disease severity compared to those with low aNE levels. High levels of active NE were correlated with low intra-patient (α) diversity, high levels of *Pseudomonas* genus, as well as high detection of *P. aeruginosa* in sputum by molecular biology; *Streptococcus, Haemophilus,* and *Staphylococcus* co-occurred instead in the low aNE group [17]. Another recent experience evaluated integrative microbiomics in bronchiectasis and researchers demonstrated the efficacy of this technique in understanding bronchiectasis exacerbations [72]. Different clusters were identified through interaction networks involving fungi, viruses and bacteria. *Pseudomonas* interactome was different based on the exacerbations rate, suggesting an association between *Pseudomonas* interactome and exacerbation risk. This experience also gave insight into exacerbations through shotgun metagenomics, confirming multi-biome interactions and the association of networks with exacerbations [72]. This approach has only been applied to multi-biome analysis, not including airway inflammation. Multi-omics approaches are nowadays focusing on chronic respiratory diseases in order to integrate different analyses such as microbiome, transcriptomic, metatranscriptomic, metabolomics, proteomic and others, to investigate host pathogen interaction, in order to identify peculiar endotypes in these patients [70].

MMPs have also been studied in the context of lung microbiota in bronchiectasis patients. MMPs quantification and 16s rRNA gene sequencing were conducted in induced sputum of 86 bronchiectasis patients and 8 healthy controls as well as clinical assessment. This experience showed increased MMP-2 and MMP-8 in patients *H. influenzae*-dominated compared to those with a *P. aeruginosa*-dominated microbiome [73].

### 5.4. Antiproteases in Bronchiectasis

In addition to proteases, also the role of antiproteases is under study in bronchiectasis. Genetic AAT deficiency has been associated with bronchiectasis, along with the association recently found between low saliva SLPI levels and bronchiectasis severity index and low sputum SLPI, exacerbation frequency and longer time to the next exacerbation [33].

#### α1 Antitrypsin Deficiency (AATD) in Bronchiectasis

Very few experiences evaluated prevalence of AATD in bronchiectasis patients. AAT serum concentration has been recently evaluated in two different cohorts from the UK: 2.5% of patients in Scotland and 7.1% in England had serum levels of AAT less than 1 g/L. Among the 1600 patients, 0.5% had PiZZ (proteinase inhibitor ZZ), 0.4% PiSZ (proteinase inhibitor SZ) or PiS (proteinase inhibitor SZ), and 3% PiMZ (proteinase inhibitor MZ) phenotype. This screening identified severe AATD in less than 1% of patients with bronchiectasis [74], confirming data from a previous study conducted in France in 2000 [75]. Another study considering PiZ patients reported 27% of these patients with clinically significant bronchiectasis. These patients showed high airway disease and more severe emphysema [76]. Although genetic deficiency of AAT is frequently associated with bronchiectasis, secondary deficiencies may also potentially contribute to bronchiectasis development.

## 6. Treatments of Protease–Antiprotease Imbalance in Bronchiectasis

Due to the central biological role of proteases in causing extracellular matrix damage, along with uncontrolled inflammation in lungs, and to the increasing evidences underlying the association between high proteases activity with disease severity and outcomes, these molecules have been identified as putative targets of treatment in bronchiectasis, as shown in Figure 1. Different strategies can be adopted to either inhibit proteases or increase antiproteases. Table 1 shows treatments that reached a clinical phase.

### 6.1. Active Neutrophil Elastase Inhibitors

Over the past decade, different molecules have been developed to target NE in bronchiectasis.

*AZD9668*. AZD9668 is an oral reversible inhibitor of NE. The most recent experience reported a small phase II double-blind controlled clinical trial on 38 bronchiectasis patients [75]. Subjects who underwent 4 weeks of treatment with this inhibitor showed an increase in pulmonary function, with a trend to a reduction of inflammatory biomarkers (IL-8- IL-6) in patients who undergo treatment vs. placebo. No difference was found in terms of sputum volume and quality of life [84]. This inhibitor was also tested in CF and COPD with inconclusive results [9].

*BAY 85–8501.* Another NE inhibitor which has been recently developed and tested in bronchiectasis is BAY 85–8501. This is a reversible and selective NE inhibitor active in animal models of acute lung injury. In phase I, no adverse effects were reported, and the drug was well tolerated at increasing dosage in healthy volunteers [85]. In a recently published phase 2a controlled randomized double-blind clinical trial, ninety-four bronchiectasis patients were treated for 4 weeks with oral BAY 85–8501. Although both tolerability and safety were proved in these patients along with a decrease in aNE in blood, no difference in terms of pulmonary function, quality of life and inflammatory and tissue damage biomarkers including desmosine in both sputum and urine were detected between patients treated with this molecule and placebo. The authors concluded that a larger study with a prolonged time treatment is needed to better evaluate clinical outcomes [77].

*CHF6333.* Different administration strategies have been adopted to increase the biodisponibility of the compounds at the target, including new molecules for NE inhibition delivered through inhalation. CHF6333 is the first inhaled candidate for NE inhibition in bronchiectasis as well as in CF. Preclinical data showed high power of the compound in inhibiting NE and data ex vivo on both bronchiectasis and CF bronchoalveolar lavage fluid samples show a higher inhibitory power of this molecule compared to BAY 85–8501. A phase I randomized clinical trial including CF and bronchiectasis patients has been concluded in 2020 (clinicaltrial.gov ID NCT04010799), although no data are currently available [86].

POL6014. POL6014 is a second inhaled compound recently proven to be safe in healthy subjects and CF patients. This molecule, safe and well tolerated in the two study groups, was reported to be successfully delivered to lungs with a poor systemic uptake [78]. A double-blind, randomized, controlled, clinical trial has been completed in 2020 in CF patients, although no data are currently available (clinicaltrial.gov ID NCT03748199).

### 6.2. DPP1 Inhibitors

While studies aimed at directly inhibiting NE in bronchiectasis patients have been carried out, a new approach aimed at targeting the enzyme responsible for serine proteases activation has been proposed (Figure 3).

*Brensocatib.* Brensocatib is an oral reversible inhibitor of DPP1, the enzyme responsible for serin proteases inhibition that has been associated with decreased serin proteases levels in healthy controls. Recently, Chalmers and colleagues published the results from a phase II, randomized, double-blind, placebo-controlled clinical trial in bronchiectasis patients [57]. Of 256 patients, 87 were assigned to receive placebo, 82 to receive 10 mg of brensocatib daily, and 87 to receive 25 mg of brensocatib for 24 weeks. Researchers found an improvement in clinical outcomes in treated vs. non-treated patients. Patients treated with two different doses of brensocatib showed an increased time to first exacerbation and a decreased incidence rate of exacerbations compared to placebo. These patients also showed a lower rate of severe exacerbations, lower negative change in FEV_1_ after bronchodilation, and a decrease in sputum aNE levels compared to the placebo ones. NE activity was restored to baseline levels after 4 weeks from the end of the drug administration [57]. However, the incidence of dental and skin adverse events was higher with both the brensocatib doses compared with placebo.

GSK2793660. Another molecule was previously developed to inhibit DPP1. Although GSK2793660 was able to inhibit the majority of DPP1, downstream serine proteases activity was inhibited in just 20% of the cases [79]. Additional studies were prematurely terminated because of adverse effects.

### 6.3. AAT Supplementation

Clinical trials aimed at investigating the efficacy of AAT supplementation reported a beneficial effect of this treatment in terms of change of pulmonary density, decreased number of exacerbations/year and improved survival [87,88]. Several trials reported increased serum levels and anti-elastase capacity of these treatments in patients with AATD, confirming the stability of AAT serum levels in patients treated with intravascular AAT supplementation without serious adverse effects along with a positive effect of supplementation therapy on clinical endpoints as lung function decline, exacerbations risk and mortality [82,88]. All treatments listed in Table 1 represent different formulation of AAT augmentation therapy developed and available on the market. All the products resulted to be biosimilar to Prolastin C and no specific data on bronchiectasis patients with AATD are available.

### 6.4. Future Perspectives

Heterogeneous results were observed in the trials studying proteases inhibitors and antiproteases activity enhancers. Although the treatments characteristics are fundamental for the mechanism of action, along with the delivery method, the selection of the correct population to test and subsequently treat with a protease inhibitor is fundamental for the success of the treatment. The development of point-of-care analysis for proteases activity detection along with an increasing knowledge of patients endotype associated with high proteases in airways is a positive step, helping both clinicians and scientists in the selection of the correct population [16,17]. The development of these tests may be very important for a quick and broad evaluation of aNE in patients and the selection of the most suitable population to treat. As mentioned, not all patients have a neutrophilic inflammation phenotype in stable state, and the heterogeneous results from clinical trials may be due to an inaccurate selection of the target population, including patients not able to respond to treatment, due to low values of aNE in lower airways. Moreover, a long-term effect at both efficacy and safety level of protease inhibition needs to be considered, and drug development is fundamental in order to achieve inflammatory inhibition without compromising host defenses, as well as inducing potential adverse effects in other departments, as previously observed [57,73].

Different situations are associated with exacerbation. These treatments have a crucial role in reducing exacerbations risk and disease progression. We previously reported the role of these molecules in decreasing exacerbation risk in patients with high levels of aNE; however, future studies should focus on the use of these compounds in inhibiting aNE in patients with low baseline levels of aNE, which usually increases during exacerbations.

The effort of the scientific community in endotyping bronchiectasis patients may pave the way to new targets of treatment and contribute to the precision medicine approach adopted for the management of this disease. Proteases and the proteases–anti-proteases imbalance represent some of the most interesting targets in bronchiectasis to be explored in future studies.

## 7. Conclusions

Protease–antiprotease balance is crucial in bronchiectasis pathophysiology. Increased presence of unopposed proteases activity may contribute to bronchiectasis onset and progression. The identification of different targets of therapy within the protease–antiprotease balance contributes to a precision medicine approach in bronchiectasis and eventually interrupts the vicious vortex, which characterizes the progression of the disease. Data from the clinical trials on DPP1 and the new trials ongoing on NE inhibitors seem very promising. However, future studies are needed to better understand the host pathogen interaction in bronchiectasis, to better characterize patients with different inflammatory patterns, to understand the role of potential treatments in use in other respiratory diseases and to identify new treatable traits in bronchiectasis.

## Figures and Tables

**Figure 1 ijms-22-05996-f001:**
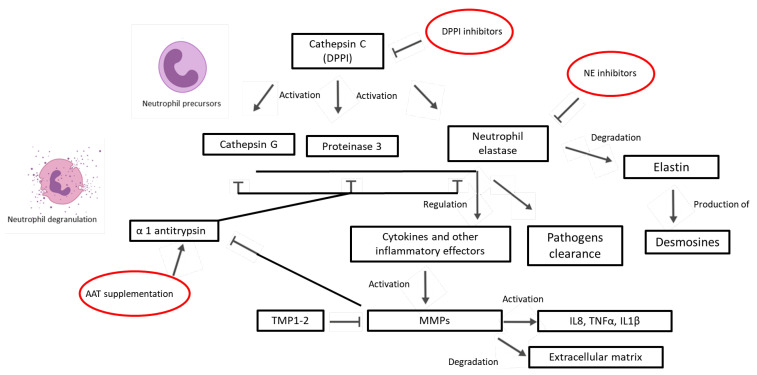
Mechanism of action of proteases and antiproteases, along with their potential treatments. DPP1: Dipeptidyl peptidase 1; AAT: α1 antitrypsin; TMPs: tissue inhibitors of metalloproteinases; MMPs: Matrix metalloproteinases; IL: interleukin; TNF: tumour necrosis factor.

**Figure 2 ijms-22-05996-f002:**
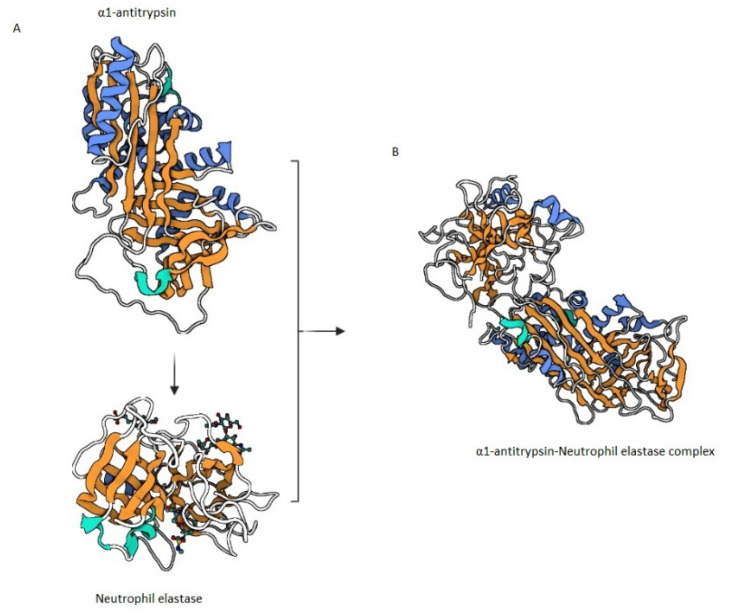
AAT mechanism of inhibition. (**A**) AAT binds NE, the reactive central loop is cleaved in a high energy state followed by a change of conformation (**B**) and the formation of an AAT-NE complex.

**Figure 3 ijms-22-05996-f003:**
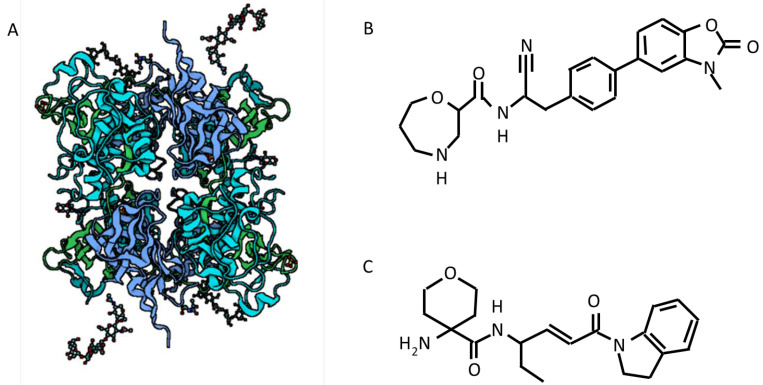
Structure of DPP1 (**A**), brensocatib (**B**) and GSK2793660 (**C**).

**Table 1 ijms-22-05996-t001:** Molecules developed to treat protease–antiprotease imbalance in bronchiectasis.

Molecule	Administration	Update	Main Evidence	Safety/Adverse Effects	References or Trial Registration
	Neutrophil elastase inhibitors				
AZD9668	Oral	Phase 2 completed	Increased pulmonary function and decreased inflammatory biomarkers (IL-6-IL-8) in patients treated vs. non treated	Safe and tolerable	[76]
BAY 85–8501	Oral	Phase 2a completed	No evidence in increasing pulmonary function and quality of life after 4 weeks’ treatment	Safe and tolerable	[77]
CHF6333	Inhaled powder	Phase 1 completed	Paper not published to date	Safe and tolerable	NCT04010799
POL6014	Inhaled	Phase 1 completed	No data available on bronchiectasis	Safe and tolerable	[78]
	Cathepsin C/DPP1 inhibitors				
GSK2793660	Oral	Phase 1	-	Terminated because of adverse events	[79]
Brensocatib	Oral	Phase 2 completed	Effective in reducing time to the first exacerbation and rate of severe exacerbations	Safe and tolerable	[57]
	AATD therapy				
Prolastin C	Intravenous	Post-marketing	Effective in increasing AAT serum levels. It reduces the decline in lung density.	Safe and tolerable	[80]
API-GLASSIA	Intravenous	Post-marketing	Bioequivalent to Prolastin-C	Safe and tolerable	[81]
Zemaira	Intravenous	Post-marketing	Dose confirmed, higher dose may be associated with greater effect. Prevents emphysema in AATD patients	Safe and tolerable	[82]
Liquid Alpha1-PI	Intravenous	Post-marketing	Bioequivalent to Prolastin-C	Safe and tolerable	[83]

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
