# Peer review of "Protease–Antiprotease Imbalance in Bronchiectasis"

_ijms, 2021, doi:10.3390/ijms22115996_

Round 1
Reviewer 1 Report
I found the work interesting, I have some suggestions
In paragraph 3.4, it would be useful to include some brief considerations on rare variants of AAT
In section 5.4, if possible add some further consideration on the potential role in respiratory exacerbations.
Author Response
I found the work interesting, I have some suggestions
Comment 1
In paragraph 3.4, it would be useful to include some brief considerations on rare variants of AAT
Answer to comment 1
We thank the reviewer for this comment, and we agree with him/her about the need to mention rare variants of AAT.
For this reason, we added at lines 220-223
“The number of rare variants represents a large variability in AAT production and functionality. These polymorphisms, sometimes with unknown function, may further be associated with deficiency with an increased number of patients who might be affected by this condition [59,60]”
Comment 2
In section 5.4, if possible add some further consideration on the potential role in respiratory exacerbations.
Answer to comment 2
We clearly understand the reviewer’s comment and we thank him/her for the suggestion. In order to improve the paragraph on respiratory exacerbations we added to lines 450-455.
“These treatments have a crucial role in reducing exacerbations risk and disease progression. We previously reported the role of these molecules in decreasing exacerbation risk in patients with high levels of aNE; however, future studies should have a focus on the use of these compounds in inhibiting aNE in patients with low baseline levels of this molecule which usually increases during exacerbations.”
Reviewer 2 Report
The review article by M. Oriano et al describes the protease/antiprotease imbalance occurring in bronchiectasis pathophysiology. The authors did a good job by reviewing mechanisms of inflammation and the main proteases and antiproteases involved together with their role in the onset and progression of the disease. Pharmacological treatments of protease/antiprotease imbalance have also been described and the reference section is quite exhaustive and appropriate.
Despite these very interesting aspects can represent a food for thought for the reader, the manuscript needs a major revision to further improve its quality and to be considered suitable for publication in IJMS.
Introduction:
Lines 43-45: Why did the authors focus their attention on bronchiectasis prevalence and incidence only in the UK? Are these data comparable to other worldwide studies? Please specify.
Line 58: a few words concerning the role of cigarette smoking in developing bronchiectasis should be introduced.
A paragraph including the methodology used for finding the articles used for the preparation of the manuscript (i.e. search methodology, study selection, data extraction and analysis) should be introduced between section 1 and 2.
Paragraph 3.4:
Since NE and AAT are the main players involved in the extracellular matrix degradation in lungs, a few sentences concerning the mechanism of NE inhibition by AAT should be reported. The authors are also kindly invited to add a figure in which a molecular model of this inhibition mechanism is shown.
Paragraph 4.2.
Line 231: a couple of references should be added after the acronym FEV1.
Paragraph 4.5
Lines 273-276: are the reported data from UK comparable to those obtained in other worldwide studies? Please specify.
Paragraph 5.1
A brief description of the NE inhibitor POI6014 should be reported.
Paragraph 5.2
A brief description of the DPPI inhibitor GSK 2793660 should be reported.
Paragraph 5.3
This paragraph should be improved by adding more information concerning the different AATD therapies.
Figures:
Figure 1: Together with the mechanisms of action of proteases and antiproteases, in Figure 1 are also reported the different types of molecules developed for treating protease/antiprotease imbalance. For this reason, this figure should be moved to section 5 before Table 1. Furthermore, as far as the figure concerns, the words “Elastase” and “Desmosine” should be replaced by “Elastin” and “Desmosines”, respectively. The sentence “Mechanisms of action of protease and antiproteases are reported in Figure 1” (lines 124-125) should be deleted in the text.
A Figure containing chemical structures of NE and DPPI inhibitors should be added in section 5.
Minor Points:
A few typographical errors are present throughout the text.
Reviewer 3 Report
The manuscript by Oriano and colleagues reviews host airway proteases and anti-proteases and their involvement in bronchiectasis, a respiratory disease characterized by a vicious cycle of infection and inflammation common to various conditions like CF and PCD. Considering the key role of protease / anti-protease imbalance in bronchiectasis, therapies targeting both sides have been developed. The manuscript is well written, but the information provided could be more detailed. Moreover, it needs careful revision since the text contains many oversights.
Major comments:
- The review lacks various elements that should be implemented to provide a comprehensive information:
- In addition to ATT, other antiproteases are SLPI (Secretory Leukocyte Protease Inhibitor) and elafin. These two are not even mentioned in the review, despite still recent research efforts to elucidate their association with bronchiectasis (e.g. Perea 2020). Some information about these two could be added. Also cystatins are never mentioned, although DPPI inhibitors are described. Are they cystatins-like?
- Similarly, cathepsin G and proteinase 3 are only mentioned in the text, but no information regarding their specific activity is presented. Also in Figure 1 the two enzymes are present but their activity is not shown. Some information about these enzymes could be added.
- Many MMPs are likely involved in bronchiectasis (e.g. Taylor 2015 found that various MMPs were significantly higher in patients with bronchiectasis than in healthy subjects), but only MMP-8 and -9 are described in this manuscript. The section regarding MMPs could be implemented to include some information about other MMPs.
- What about bacterial proteases? Bacterial infection is a component of bronchiectasis, and proteases secreted by bacteria may be involved in the disease too, as outlined in literature (e.g. Travis 2000; Kipnis 2006; Guyot 2010; Sandri 2018). This should be at least mentioned in the manuscript. If the authors prefer to focus the review on host proteases only, this should be stated in the manuscript.
- Only treatments that reached clinical experimentation are described. The authors could add information (maybe in a new section) about molecules that have demonstrated efficacy in vitro and in the preclinical in vivo setting too, especially those regarding different mechanisms/targets (e.g. inhibition of proteases other than NE and CatC).
- Future perspectives section could be expanded with discussion about the possibility of targeting other proteases (human and/or bacterial) or more than one (larger spectrum inhibitors), supplementing other anti-proteases, etc.
- Almost all references cited in section 3.2 (about MMPs) are quite old and should be update with more recent publications when possible.
Minor comments:
There are many oversights in the text, please carefully check the whole manuscript.
- Abstract, line 21: over-reactivity
- Abstract, line 27: PiSZ and PiZZ acronyms should be explained.
- Introduction, lines 52-54: add one sentence of information about cystic fibrosis too. Later in the text cystic fibrosis is mentioned but was never introduced.
- Lines 77, 79: IL-8, IL-1beta require hyphen. Check it throughout the whole manuscript, also for other interleukins.
- Lines 77-78: "Imbalances in cytokines interplay in bronchiectasis airways are associated".
- Line 85: “In bronchiectasis patients”. In which sample / site? E.g. in sputum / lung tissue. Please specify.
- Lines 89-92: Is the eosinophilic immune response additional or alternative to neutrophilic inflammation? Does it involve proteases too? Please add some details regarding these aspects.
- Line 105: Neutrophil serine proteases
- Lines 110-111: “cathepsin C, also known as dipeptidyl peptidase 1 (DPP1)”
- Line 116: and are able
- Line 120: “These proteins…”. "Enzymes" would be more appropriate.
- Line 120: “also have a role in the activation/inactivation of cytokines”. This was already stated in the previous sentence (lines 117-118). I suggest to delete it from the previous sentence and leave it only here.
- Line 124: "other" inflammatory effectors, since interleukins were already mentioned in the previous sentence.
- Lines 124-125: Move this sentence and the figure before the "serine proteases" section, since it is general and does not regard specifically these enzymes.
- Figure 1:
- It looks like CatG and PR3 do nothing in terms of inflammatory effectors activation. Add their activity.
- NE degrades elastin (not elastase). Please correct.
- MMP-1 is never cited in the text, MMP-2 only once. If they are not key MMPs like MMP-8 and -9, remove them from the figure.
- Line 131: ELANE acronym should be explained.
- Line 140: “interleukin-6 (IL-6)”. No need for the extended form, IL acronym was already explained in line 76. Writing IL-6 is enough.
- Lines 146-149: this sentence does not regard NE specifically, but is a general consideration about proteases. I suggest removing it or moving it to another section.
- Line 151 and 153: MMPs. Check it throughout the whole manuscript.
- Line 171: “Cathepsin C, also known as dipeptidyl peptidase I (DPPI)”. No need for the extended form, the acronym was already explained in line 110. Writing DPPI is enough.
- Line 175: serine
- Lines 176-177: “molecules were developed and tested in patients with bronchiectasis in order to inhibit NE activity in bronchiectasis”.
- Line 184: SERPINA1 acronym should be explained.
- Line 187: “the most common variants associated with AAT serum deficiency are Z and S allele (PiZZ and PiSZ)”. Add the acronyms.
- Line 195: “maybe because of an excess in chemoattractants”. Add reference.
- Line 195: “lack of AAT”
- Line 197: “The role of proteases and antiproteases in bronchiectasis”
- Line 206 and 210-211: substitute “active neutrophil elastase” with “aNE” (acronym already used). Check it throughout the whole manuscript (lines 249-250, etc.).
- Lines 220-221: Pseudomonas should be written in italic and be followed by “sp.” if you mean only one species or “spp.” for several species (plural). Check this rule throughout the whole manuscript, also for other bacteria (lines 251-252, 256, 268-269, etc).
- Line 227: Desmosine is mentioned here for the first time. Since it is not a protease, nor an anti-protease but a proteolytic product of NE, this section could be merged with the previous one (NE in bronchiectasis). Moreover, I suggest first introducing desmosine in section 3.1.1
- Line 231: FEV1 acronym should be explained.
- Line 234: cardiovascular death
- Line 237: “MMPs”, “MMPs-TIMPs”
- Line 239: ratio
- Line 243: between high levels of MMPs and poor lung function
- Line 244: This section contains studies regarding both microbiota and microbiome, it’s not correct to cite only one of them in the section title. Add “microbiota” in the section title or reformulate the title to include both.
- Line 251: “…in sputum by molecular biology”
- Line 258: an insight on exacerbations
- Line 265: substitute “microbiome” with “microbiota”.
- Line 268: “This experience showed increased MMP-2 and MMP-8 increased in patients…”
- Line 274: PiMZ acronym should be explained
- Table 1:
- I suggest dividing the “main evidence” column into 2 sub-columns, one regarding the effect in treating bronchiectasis and the other regarding safety/collateral effects, so that both information are more immediate for the reader.
- Line GSK2793660, “terminated due to adverse events” is written twice. Delete one.
- Lines API-GLASSIA and liquid A1-pha1-P1: “bioequivalent to prolastin-C”. Are the effects on bronchiectasis also similar? Please add details to clarify.
- Line 292: AZD9668 is an oral…
- Line 316: data ex vivo on both bronchiectasis and CF bronchoalveolar lavage fluid samples show a higher inhibitory power…
- Line 317: Use CF instead of cystic fibrosis (acronym already explained)
- Line 318: “recently”. When? Add the year.
- Line 319: What about POL6014? It's present in the table but not even mentioned in the text. Please add the relevant information.
- Line 328: “10 mg of brensocatib”. Daily? Please clarify.
Author Response
The manuscript by Oriano and colleagues reviews host airway proteases and anti-proteases and their involvement in bronchiectasis, a respiratory disease characterized by a vicious cycle of infection and inflammation common to various conditions like CF and PCD. Considering the key role of protease / anti-protease imbalance in bronchiectasis, therapies targeting both sides have been developed. The manuscript is well written, but the information provided could be more detailed. Moreover, it needs careful revision since the text contains many oversights.
The review lacks various elements that should be implemented to provide a comprehensive information:
Major comment 1
In addition to ATT, other antiproteases are SLPI (Secretory Leukocyte Protease Inhibitor) and elafin. These two are not even mentioned in the review, despite still recent research efforts to elucidate their association with bronchiectasis (e.g. Perea 2020). Some information about these two could be added. Also cystatins are never mentioned, although DPPI inhibitors are described. Are they cystatins-like?
Answer to major comment 1
We thank the reviewer for his/her suggestion and we agree that a mention to SPLI and elafin is needed. For this reason we decided to modify antiproteases section accordingly.
We added at lines 238-248 and 331-335.
“4.4.2 Antileukoprotease (ALP) superfamily
Antileukoprotease (ALP) superfamily includes enzymes secreted in the lung that show an inhibitory effect on airway proteases. These proteins are synthesized and se-creted locally in the lung in response to primary cytokines such as IL-1 and TNF [62]. This superfamily includes elafin and secretory leukocyte proteinase inhibitor (SLPI).
SLPI. SLPI inhibits NE, Cat-G, trypsin, chymotrypsin and chymase, with a high af-finity to NE that is its major target. SLPI is believed to have an anti-inflammatory and anti-bacterial role [62].
Elafin. Elafin is a protein which inhibits a more limited number of proteases in comparison to SLPI. Its main targets are NE and PR3. In addition to its inhibitory ef-fect, elafin seems to be active against P. aeruginosa [62].”
“5.4. Antiproteases in bronchiectasis
In addition to proteases, also the role of antiproteases role in bronchiectasis is under study in bronchiectasis. Genetic AAT deficiency has been associated with bronchiecta-sis, along with the association recently found association between low saliva SLPI lev-els and bronchiectasis severity index and low sputum SLPI, exacerbation frequency and longer time to the next exacerbation [32].”
Major comment 2
Similarly, cathepsin G and proteinase 3 are only mentioned in the text, but no information regarding their specific activity is presented. Also in Figure 1 the two enzymes are present but their activity is not shown. Some information about these enzymes could be added.
Answer to major comment 2
We thank the reviewer for his/her suggestion about Cat-G and PR3. We decided to add at lines 145-151 the following paragraph and to modify the figure accordingly.
“Cat-G clears pathogens, modifies chemokine and cytokines and, thus, regulates inflammation [43]. Interestingly, Cat-G have been associated to poor P. aeruginosa clearance in CF mouse airways [44].
PR3 is an enzyme able to cleave structural proteins for tissue remodelling, to regulate immune response to bacterial triggers through cleavage of antibacterial peptides, activation of pro inflammatory cytokines and regulation of cellular processes. PR3 is also active in cleaving C1 inhibitor, IL-8, TNF, IL-1, TGF [45].”
Major comment 3
Many MMPs are likely involved in bronchiectasis (e.g. Taylor 2015 found that various MMPs were significantly higher in patients with bronchiectasis than in healthy subjects), but only MMP-8 and -9 are described in this manuscript. The section regarding MMPs could be implemented to include some information about other MMPs.
Answer to major comment 3
We thank the reviewer for the comment and we agree with the need of adding information about the other MMPs.
For this reason we decided to add at line 191-195.
“Several MMPs are associated with bronchiectasis. Taylor and colleagues reported increased values of MMP-1, MMP-3, MMP-7, MMP -8, and MMP -9 and TIMP-2 and -4 levels, as well as MMP-8/TIMP-1 and MMP-9/TIMP-1 ratios in patients vs. healthy controls: Moreover, MMP-2 and MMP-8 are increased in patients with H. influenzae dominated microbiota compared to those dominated by P. aeruginosa [69]”
Major comment 4
What about bacterial proteases? Bacterial infection is a component of bronchiectasis, and proteases secreted by bacteria may be involved in the disease too, as outlined in literature (e.g. Travis 2000; Kipnis 2006; Guyot 2010; Sandri 2018). This should be at least mentioned in the manuscript. If the authors prefer to focus the review on host proteases only, this should be stated in the manuscript.
Answer to comment 4
We clearly understand the reviewers point and we thank him/her for the comment. We agree that bacterial proteases may be involved in this disease, however we decided to focus on host proteases. For this reason we decided to add “Although also bacteria may produce proteases that may be involved in the protease-antiproteases balance in bronchiectasis, we decided to focus only on proteases produced by the host [22-23].” at line 70-72
Major comment 5
Only treatments that reached clinical experimentation are described. The authors could add information (maybe in a new section) about molecules that have demonstrated efficacy in vitro and in the preclinical in vivo setting too, especially those regarding different mechanisms/targets (e.g. inhibition of proteases other than NE and CatC).
Answer to major comment 5
We thank the reviewer for this suggestion and we clearly understand his/her concern about the treatments included. In order make clear to the reader that we just included treatments that reached clinical phases, we decided to add at line 355
“Different strategies can be adopted to either inhibit proteases or increase antiproteases. Table 1 shows treatments that reached a clinical phase”
Major comment 6
Future perspectives section could be expanded with discussion about the possibility of targeting other proteases (human and/or bacterial) or more than one (larger spectrum inhibitors), supplementing other anti-proteases, etc.
Answer to major comment 6
We thank the reviewer for his/her comment and we clearly understand the need of underlying the possibility to target other proteases, more than one or supplement other anti-proteases.
We decided to add to the manuscript at lines 456-460.
“The effort of the scientific community in endotyping bronchiectasis patients may pave the way to new targets of treatment and contributec to the precision medicine approach adopted for the management of this disease. Proteases and the proteases-anti proteases unbalance represent some of the most interesting targets in bronchiectasis to be explored in future studies.”
Major comment 7
Almost all references cited in section 3.2 (about MMPs) are quite old and should be update with more recent publications when possible.
Answer to major comment 7
We thank the reviewer for this comment and we provided newer references whenever possible.
Minor comments:
There are many oversights in the text, please carefully check the whole manuscript.
- Abstract, line 21: over-reactivity
- Abstract, line 27: PiSZ and PiZZ acronyms should be explained.
- Introduction, lines 52-54: add one sentence of information about cystic fibrosis too. Later in the text cystic fibrosis is mentioned but was never introduced.
- Lines 77, 79: IL-8, IL-1beta require hyphen. Check it throughout the whole manuscript, also for other interleukins.
- Lines 77-78: "Imbalances in cytokines interplay in bronchiectasis airways are associated".
- Line 85: “In bronchiectasis patients”. In which sample / site? E.g. in sputum / lung tissue. Please specify.
- Lines 89-92: Is the eosinophilic immune response additional or alternative to neutrophilic inflammation? Does it involve proteases too? Please add some details regarding these aspects.
- Line 105: Neutrophil serine proteases
- Lines 110-111: “cathepsin C, also known as dipeptidyl peptidase 1 (DPP1)”
- Line 116: and are able
- Line 120: “These proteins…”. "Enzymes" would be more appropriate.
- Line 120: “also have a role in the activation/inactivation of cytokines”. This was already stated in the previous sentence (lines 117-118). I suggest to delete it from the previous sentence and leave it only here.
- Line 124: "other" inflammatory effectors, since interleukins were already mentioned in the previous sentence.
- Lines 124-125: Move this sentence and the figure before the "serine proteases" section, since it is general and does not regard specifically these enzymes.
Figure 1:
- It looks like CatG and PR3 do nothing in terms of inflammatory effectors activation. Add their activity.
- NE degrades elastin (not elastase). Please correct.
- MMP-1 is never cited in the text, MMP-2 only once. If they are not key MMPs like MMP-8 and -9, remove them from the figure.
- Line 131: ELANE acronym should be explained.
- Line 140: “interleukin-6 (IL-6)”. No need for the extended form, IL acronym was already explained in line 76. Writing IL-6 is enough.
- Lines 146-149: this sentence does not regard NE specifically, but is a general consideration about proteases. I suggest removing it or moving it to another section.
- Line 151 and 153: MMPs. Check it throughout the whole manuscript.
- Line 171: “Cathepsin C, also known as dipeptidyl peptidase I (DPPI)”. No need for the extended form, the acronym was already explained in line 110. Writing DPPI is enough.
- Line 175: serine
- Lines 176-177: “molecules were developed and tested in patients with bronchiectasis in order to inhibit NE activity in bronchiectasis”.
- Line 184: SERPINA1 acronym should be explained.
- Line 187: “the most common variants associated with AAT serum deficiency are Z and S allele (PiZZ and PiSZ)”. Add the acronyms.
- Line 195: “maybe because of an excess in chemoattractants”. Add reference.
- Line 195: “lack of AAT”
- Line 197: “The role of proteases and antiproteases in bronchiectasis”
- Line 206 and 210-211: substitute “active neutrophil elastase” with “aNE” (acronym already used). Check it throughout the whole manuscript (lines 249-250, etc.).
- Lines 220-221: Pseudomonas should be written in italic and be followed by “sp.” if you mean only one species or “spp.” for several species (plural). Check this rule throughout the whole manuscript, also for other bacteria (lines 251-252, 256, 268-269, etc).
- Line 227: Desmosine is mentioned here for the first time. Since it is not a protease, nor an anti-protease but a proteolytic product of NE, this section could be merged with the previous one (NE in bronchiectasis). Moreover, I suggest first introducing desmosine in section 3.1.1
- Line 231: FEV1 acronym should be explained.
- Line 234: cardiovascular death
- Line 237: “MMPs”, “MMPs-TIMPs”
- Line 239: ratio
- Line 243: between high levels of MMPs and poor lung function
- Line 244: This section contains studies regarding both microbiota and microbiome, it’s not correct to cite only one of them in the section title. Add “microbiota” in the section title or reformulate the title to include both.
- Line 251: “…in sputum by molecular biology”
- Line 258: an insight on exacerbations
- Line 265: substitute “microbiome” with “microbiota”.
- Line 268: “This experience showed increased MMP-2 and MMP-8 increasedin patients…”
- Line 274: PiMZ acronym should be explained
- Table 1:
- I suggest dividing the “main evidence” column into 2 sub-columns, one regarding the effect in treating bronchiectasis and the other regarding safety/collateral effects, so that both information are more immediate for the reader.
- Line GSK2793660, “terminated due to adverse events” is written twice. Delete one.
- Lines API-GLASSIA and liquid A1-pha1-P1: “bioequivalent to prolastin-C”. Are the effects on bronchiectasis also similar? Please add details to clarify.
- Line 292: AZD9668 is an oral…
- Line 316: data ex vivo on both bronchiectasis and CF bronchoalveolar lavage fluid samples show a higher inhibitory power…
- Line 317: Use CF instead of cystic fibrosis (acronym already explained)
- Line 318: “recently”. When? Add the year.
- Line 319: What about POL6014? It's present in the table but not even mentioned in the text. Please add the relevant information.
- Line 328: “10 mg of brensocatib”. Daily? Please clarify
Answer to minor comments
We thank the reviewer for this accurate revision. We modified the text as suggested.
Specific response to some of the points is provided.
Minor comment 3
Introduction, lines 52-54: add one sentence of information about cystic fibrosis too. Later in the text cystic fibrosis is mentioned but was never introduced.
Response to minor comment 3
Although CF is citated in the paper along with other chronic respiratory diseases, it is not central in this review. For this reason, we decided to add at line 39 that we are specifically talking about non-CF bronchiectasis.
“Non-cystic fibrosis bronchiectasis (from now on reported as bronchiectasis) is a chronic respiratory disease characterized by an irreversible pathological dilation of the bronchi associated with a chronic syndrome of cough, sputum production and recurrent respiratory infections [1].”
Minor comment 7
Lines 89-92: Is the eosinophilic immune response additional or alternative to neutrophilic inflammation? Does it involve proteases too? Please add some details regarding these aspects.
Response to minor comment 7
We thank the reviewer for this suggestion. We are citating eosinophilic inflammation in bronchiectasis because is a very relevant, thus unknown field of study. Very few papers with a large heterogeneity in the definition of eosinophilic inflammation have been published. For this reason we decided to add at line 108-110 the following sentence.
“Proteases may also have a role in the eosinophilic response and further studies will be needed to unravel this aspect of bronchiectasis pathophysiology”
Minor comment 47
Lines API-GLASSIA and liquid A1-pha1-P1: “bioequivalent to prolastin-C”. Are the effects on bronchiectasis also similar? Please add details to clarify.
Response to minor comment 47
We thank the reviewer for this suggestion and we agree that a further specification is needed, although AATD is a rare disease and no data on the effects of this bioequivalent product on bronchiectasis are available in literature.
Round 2
Reviewer 2 Report
The authors have replied to all queries raised by the reviewer.
Author Response
Dear Prof. Dr. Martin
The authors would like to thank the editor for giving them the opportunity to submit a revised version of the paper and the reviewer for his/her accurate revision of the manuscript
Reviewer 3 Report
Thanks for the careful revision, the manuscript is much improved.
I’m satisfied with all the authors’ responses but one (response to minor comment):
“Although CF is citated in the paper along with other chronic respiratory diseases, it is not central in this review. For this reason, we decided to add at line 39 that we are specifically talking about non-CF bronchiectasis”.
I agree that CF is not central here; however, it is cited many times in the manuscript and studies regarding treatments tested in CF patients are here reported (some are tested even only in CF patients). Thus, it is not correct to state that the review is about non-CF bronchiectasis. From what I understood, the review is about bronchiectasis in general, including CF bronchiectasis too; otherwise, you wouldn’t need to mention CF-related studies throughout the text. I suggest to revert the initial sentence of the manuscript (line 39) to its previous form and to add at least one information about CF in the introduction. For example, in lines 56-58, you can just add CF along with PCD and add a reference about CF: “Ciliary dysfunction and failure of the muco-ciliary clearance, as demonstrated in patients with primary ciliary dyskinesia (PCD) and cystic fibrosis (CF), increases the risk of pulmonary infections and airway inflammation leading to the chronicity of the vortex.”
There are still some minor revisions needed:
- Abstract, line 27: PiZZ acronym is still missing. Please substitute “as PiSZ (proteinase inhibitor SZ) and PiZZ phenotype” with “as PiSZ and PiZZ (proteinase inhibitor SZ and ZZ) phenotype, …”. Do the same in lines 341-342 for all acronyms.
- Line 145: chemochines (plural)
- Line 146: “Interestingly, Cat-G have has been associated…”
- Line 147: this is the first time you mention CF, change in “cystic fibrosis (CF)” and remove the acronym explaination in line 206.
- Line 150: pro-inflammatory
- Line 332-334: “…also the role of antiproteases role in bronchiectasis is under study in bronchiectasis. Genetic AAT deficiency has been associated with bronchiectasis, along with the association recently found association between…”
- Table 1, line GSK2793660: in the previous version of the manuscript, it was written “terminated because of adverse events”, now it is “safe and tolerable”. Quite different outcomes. Which one is correct? Please check
- DPP1 is sometimes written DPPI. Please choose one form and be consistent in the text.
- Line 213: serine proteases (twice in the same line)
- Line 458: contibutec
Author Response
Dear Prof. Dr. Martin
The authors would like to thank the editor for giving them the opportunity to submit a revised version of the paper. We agree with all the comments and recommendations suggested by the reviewers. We have changed the manuscript to comply with reviewers’ recommendations.
Reviewer 3
We clearly understand the reviewers view and we agree to modify the text to include all bronchiectasis (CF and non-CF) as suggested.
We addressed all the minor correction the reviewer reported and we thank him/her for the precise revision of the manuscript.